# Willingness to engage in and current status of social participation among Chinese merchant sailors

**Huarong Wang**[⊙], **Yuheng He**[⊙], **Licheng Shi**, **Jiali Wang**, **Lvqing Miao**, **Jiajun Dai***

Department of Traffic Psychology, Institute of Special Environmental Medicine, Nantong University, Nantong, Jiangsu Province, China

⊙ These authors contributed equally to this work.
* yeluo801004@ntu.edu.cn

## Abstract

China has the largest population of sailors in the world, but little is known of their social participation. This study examined Chinese merchant sailors' social participation using a nationwide survey. Across 12 Chinese provinces, 7,296 merchant sailors completed the questionnaire on sailor' willingness to engage in and status of social participation. The results showed that most Chinese merchant sailors were willing to participate in social affairs, but few of them reported having joined relevant social organizations, over half of sailors reported never having participated in public affairs, and half of them chose to ignore when they faced with an obvious mistake in shipping-related information in the media. Most of sailors reported unknowing the role of the labor union related to Chinese seafarers and NGO related to navigation well, and their evaluation of these organizations were mostly negative. Chinese merchant sailors reported higher expectations of services in terms of protection of rights, providing information and technology, and providing employment opportunity. We conclude that Chinese merchant sailors have willingness to social participation although the reality is not positive and discuss implications for improving the social participation of Chinese merchant sailors.

## 1. Introduction

A sailor's life, especially that of the merchant sailor, is characterized by continual departures from and returns to the family and community. Being onboard ship effectively cuts off land-bound social contact and most shore-side activities. It reduces opportunities to establish and maintain meaningful social relationships [1, 2] and participate in social affairs. Oldenburg and colleagues [3] think that long periods of separation from the family is a main stress parameter for sailors at sea. Because the sailors suffer from the absence of routine family relationships and community ties, they may be seen as immature, carefree, and lacking in social skill when ashore [4]. However, little is known of what willingness merchant sailors have to participate socially or the status of their social participation.

**Data Availability Statement:** All relevant data are within the manuscript and its Supporting Information files.

**Funding:** This work was supported by the China Association for Science and Technology Research

Project [grant number 2015DCYJ04] and Large Instruments Open Foundation of Nantong University(2020).The funders had no role in study design, data collection and analysis, decision to publish, or preparation of the manuscript.

**Competing interests:** The authors have declared that no competing interests exist.

Social participation improves health and is an important component for quality of life [5]. It is also the most common measure for social capital, which is associated with well-being [6–8]and psychological health [9, 10]. Governments and other organizations work to promote it. For example, the encouragement to support age-friendly communities by World Health Organization [11] hopes to prompt governments to enable their elderly to participate more in social activities and improve their health by that means.

In China, social participation is considered to be a basic part of Chinese social construction [12], but China's civil society is still in the process of development and formation due to the basic characteristics of "strong country and weak society" in the traditional Chinese society, which basically eliminate the existence of civil society [13]. Therefore, many efforts from all walks of life to promote the social participation of citizens, and various forms of social participation are in the ascendant in China. For example, Chinese government issued the first youth development plan, the medium and long term youth development plan (2016–2025) in 2017, which clearly stated that " further enrich and unblock the channels and ways of youth social participation, and realize the active, orderly, rational and legal participation of youth groups in the socialist modern construction" [14]. China now boast the world's largest Internet population of 854 million [15], and online participation has become a common method for Chinese citizens to contact political authority and articulate their issues and grievances [16, 17]. Moreover, researchers have paid more and more attention to social participation of some specific disadvantaged groups, including older adults [18, 19], disabled groups [20, 21], and migrant workers [22, 23] et al. Similar to the findings of other countries, many researches confirm that social participation contributes to Chinese health [24], including mental health [25, 26], and political trust [27]. However, as another relatively vulnerable group (due to the separateness of their onboard and shorebound lives), merchant sailors' social participation has been seldom examined and is little known in the Chinese context, although China has the largest population of registered sailors in the world [28].

This study was conducted to extend the literature on social participation among Chinese merchant sailors. Because of the lack of previous evidence, we designed an exploratory study, examining three aspects: willingness to engage in social participation, the state of social participation, and the evaluation of and expectations for the current environment for social participation.

## 2. Methods

### 2.1 Participants

A nationwide cross-sectional study was conducted in 2015. The survey was conducted among 13 national maritime bureaus spread across 12 Chinese provinces to take account of the social participation of merchant sailors in relation to all parts of the country. In all, 10,000 merchant sailors were recruited from 12 provinces. This sample yielded 7,296 valid questionnaires (effective response rate of 72.96%), representing 2,386 (32.7%) sailors aged less than 30 years old, 2416 (33.11%) aged 30–39 years old, 1527 (20.93%) aged 40–49 years old, and 625 (8.57%) aged 50 years old above, in addition to 342 responses with unknown age (4.69%). All of the merchant sailors were male. The full age range was 18 to 67 years old, with an average age of 35.09 ($SD$ = 9.34) years. Other demographic details on the participants are given in Table 1.

### 2.2 Questionnaire

The data were collected using a questionnaire that was developed from a literature review and interviews with sailors and personnel managers working in maritime transportation. The questionnaire included items on the willingness to engage in, current status of, and evaluation

**Table 1. Demographic information on the participants.**

| Variables | N | % | Variables | N | % |
|---|---|---|---|---|---|
| *Educational background* | | | *Length of career* | | |
| Bachelor degree and above | 1322 | 18.12 | 5 years or less | 1842 | 25.2 |
| Junior college diploma | 3630 | 49.76 | 6–10 years | 2239 | 30.7 |
| High school diploma | 1581 | 21.67 | 11–20 years | 1653 | 22.7 |
| Junior high school diploma and below | 410 | 5.62 | 21years or more | 1369 | 18.8 |
| *Residence* | | | *Types of contract* | | |
| City | 1396 | 19.14 | Contract worker | 4073 | 55.83 |
| Small urban area | 1624 | 22.26 | Dispatched workers | 1170 | 16.04 |
| Rural | 3813 | 52.27 | Individual crew | 1828 | 25.06 |
| *Types of ships working* | | | *Sailing area* | | |
| Ordinary cargo ship | 4801 | 65.81 | Ocean-going area | 4003 | 54.87 |
| Oil and Gas Ships | 2104 | 28.84 | Coastal area | 3068 | 42.05 |

Note: Information was unreported for the educational background of 353 sailors, for the residence of 463, the types of ships working of 391, the sailing area of 225, the working age of 193, and the type of contract for 225.

of social participation. The validity of the questions assessing merchant sailor social participation was confirmed by experts. A pilot study with 235 sailors was conducted to establish the clarity of the question items, and the questionnaire was revised in light of the responses to this trial. The revised questionnaire, which consisted of 19 items, was used in the main study.

Two items reflected willingness to engage in social participation, including the concern for national policies, as well as the willingness to participate in public affairs (e.g., "Would you like to participate in public affairs in your community?", 1 = very willing, 2 = willing, 3 = unwilling, 4 = completely unwilling, 5 = not sure). Eleven items investigated the current status of the respondent's social participation from three aspects, that is, participation in politics (e.g. "Are you, or were you, a deputy to the National People's Congress at any levels?, 1 = yes, 2 = no), participation in public affairs beyond politics (e.g., "Have you given comments or suggestions to community leaders?", 1 = often, 2 = sometimes, 3 = no), and joining social organizations (e.g. "Are you a member of the labor union related to Chinese seafarers", 1 = yes, 2 = no). Six items assessed the evaluation of social participation (e.g., "Do you think the channels for participating in public affairs for sailors are smooth at present?" 1 = very smooth, 2 = smooth, 3 = not very smooth, 4 = not smooth at all, 5 = not sure).

The questionnaire also prompted participants to report demographic information (age, residence, length of career, sailing area, educational background, types of ships worked on, and type of contract; see Table 1).

## 2.3 Procedures

Responses were collected from 7,296 merchant sailors from 13 national maritime bureaus across 12 Chinese provinces, reached cluster sampling. This investigation was supported by the Maritime Safety Administration of the People's Republic of China, which arranged for its 13 local maritime safety bureaus to assist in organizing local shipping companies to participate in the investigation. Each sailor participating in the research was informed in detail of the purpose and requirements of the research before the survey. After the sailors verbally agreed to participate in the research, two research assistants sent questionnaires to merchant sailors and helped them fill them out, while those who did not agree to participate in the survey would not fill out the questionnaire. Approval for the research was obtained from the Nantong University

Academic Ethics Committee prior to the study (Tong Da Lun Shen (2015) No.60). Participants were guaranteed anonymity and that their answers would remain confidential. The study took about 10 to 15 minutes to complete for each participant.

## 3. Results

Descriptive analyses and $\chi^2$ tests were used to assess the results. Preliminary analyses examined the relationships between sailors' demographic variables and their responses to series questions; all demographic variables, with the exception of age group, were omitted from subsequent analyses due to their null results in the preliminary analyses.

### 3.1 Willingness to engage in social participation

We focused on merchant sailors' concern for national policies, as well as their willingness to participate in public affairs. Most respondents (66.3% of the sample) expressed great concern or concern regarding national policies, while 30% of the sample reported that they were "no concern" or "little concern", and 3.7% of them reported "not sure". A $\chi^2$ test produced a significant difference among the three groups (we combined values for "great concern" and "concern" as a group, and combined values for "no concern" or "less concern" as a group) of respondents, $\chi^2$ (2, $N = 6694$) = 13.5, $p < 0.05$, and the proportion of merchant sailors who expressed complete disregard for national policies was significantly less than the group of sailors who expressed concern.

We examined concern in relation to national policies across age groups. A slight difference was found in attention to national policies, which showed the trend that sailors over 40 years old paid more attention to national policies than those under the age of 40 (Fig 1).

In response to "Would you like to participation in public affairs in your community?", 14.2% of the merchant sailors answered "I'm very willing" and 53.4% of them reported "I'm willing", 20.7% responded that they were "unwilling", and 1.9% merchant sailors answered

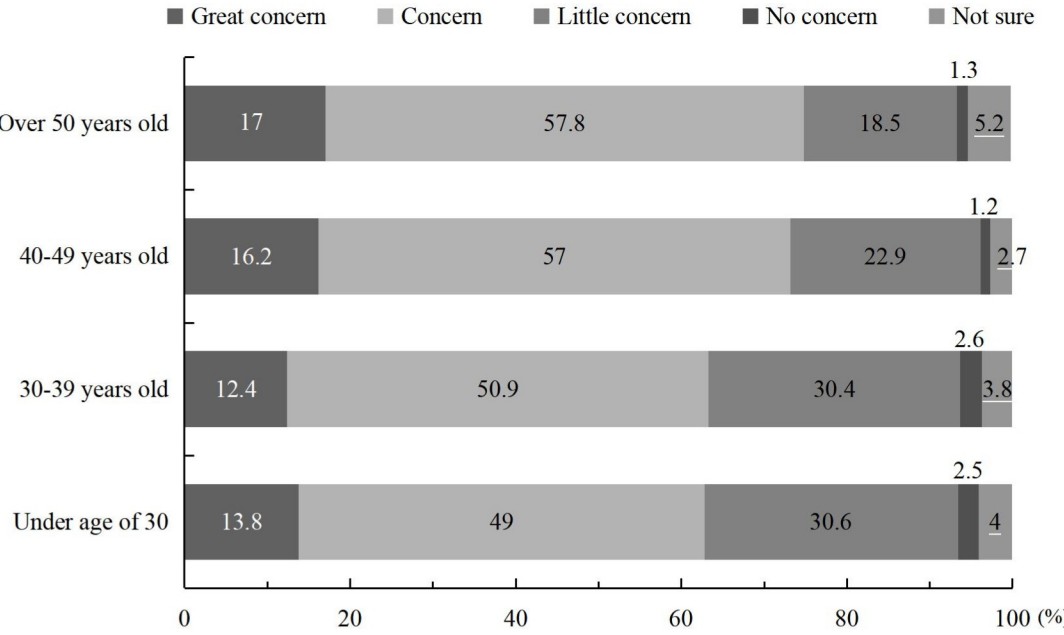

**Fig 1. Concern for national policies across age groups of Chinese merchant sailors (%).**

that they were "completely unwilling", with the remaining 9.8% of the sailors reporting "I'm not sure". A $\chi^2$ test suggested that the difference among the responses was significant ($\chi^2$ (4, $N = 6694$) = 55.03, $p < 0.01$), and the positive option "willing", received significantly more of the response than the others.

The merchant sailors' willingness to participate in the public affairs differed across age groups, as shown in Fig 2. Those who were under 30 years old tended to express more willingness to participate in public affairs (we combined values for "I'm very willing" and "I'm willing" as a group, and combined values for "I'm unwilling" or "I'm completely unwilling" as a group).

## 3.2 Social participation

We investigated the sailors' social participation in relation to three aspects, namely, participation in politics, participation in public affairs beyond politics, and joining social organizations. This showed that 2.9% of merchant sailors are deputies to the National People's Congress, and 1.8% are members of the Chinese People's Political Consultative Conference (CPPCC), a similar participation ratio to that of Chinese scientific and technical workers (of whom 1.7% are deputies to the National People's Congress, and 1.7% are members of the CPPCC).

We examined the proportion of sailors who were members of the labor union related to Chinese seafarers, the non-governmental organizations (NGO) related to navigation in China. In our sample, 19.5% were members of the labor union related to Chinese seafarers, and 12.6% were members of NGO related to navigation.

We also investigated sailors' participation in a range of public affairs. As is shown in Fig 3, 60.8% of sailors reported participating in public affairs by "making comments to people around you" (combined values for "often" and "sometimes" answers), 53.5% contributing

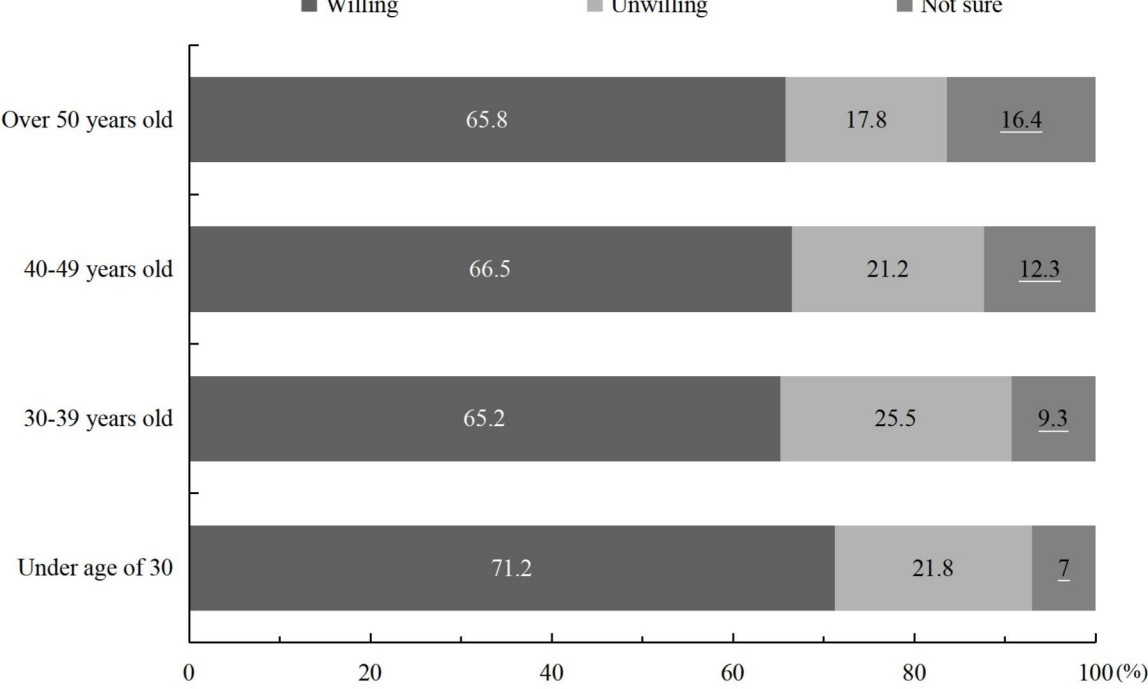

**Fig 2. Willingness to participate in public affairs among Chinese merchant sailors, broken down by age group (%).**

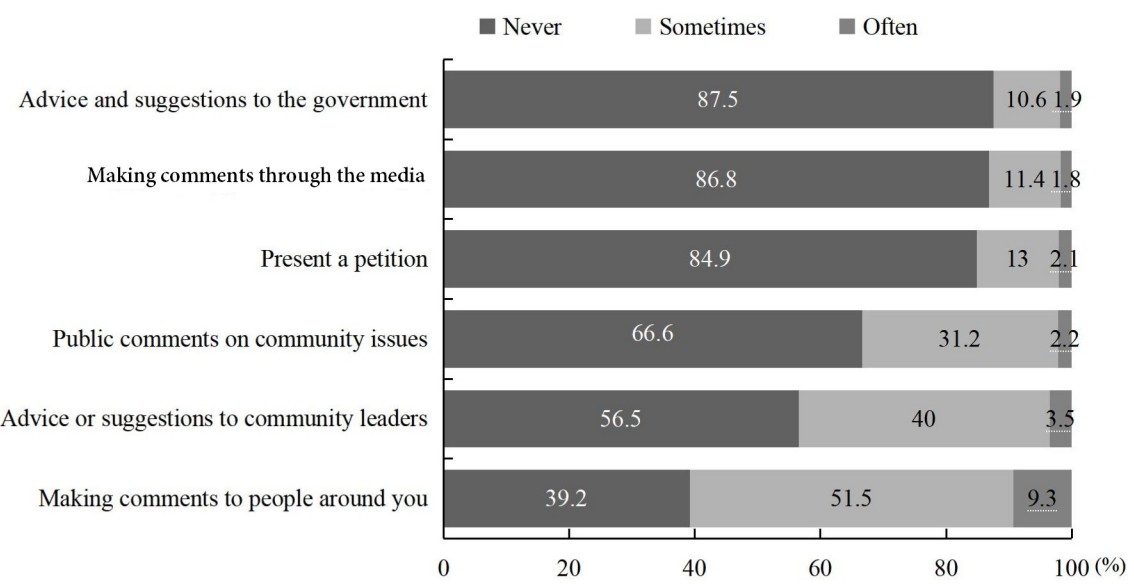

**Fig 3. The engagement in public affairs that Chinese merchant sailors participate in (%).**

"advice or suggestions to community leaders", and 33.4% giving "public comments on community issues". Only 12.5% gave "advice and suggestions to the government".

Slight differences were found in participation in public affairs across age groups (Table 2). Specifically, more sailors 40 years old or older who reported caring about the public affairs of their communities than under 40 years old (51.3% reported caring in among those 50 years old and older, 49.4% in among those 40–49 years old, opposed to 41.9% among those 30–39 years old and 39.6% among those under 30 years old), and the group under 40 years old reported a greater likelihood of making comments through the news media or to the government than the group 40 years old or older (e.g., the proportions of reporting "making suggestions through the media" across age groups was 17.5% and 12.2% vs. 8.7% and 9.5%, respectively).

We examined merchant sailors' response to obvious mistakes in shipping-related information or reports in the media. Most respondents (50.2%) said they would not mention the erroneous information, and 30.8% reported that they would "clarify the mistake through social media such as QQ and Wechat", with only 10.9% of reporting that they would "contact the media directly and point out the error". Fig 4 listed a detailed response to incorrect

**Table 2. Chinese merchant sailors' participation in a range of public affairs across age groups.**

|  | Public comments on community issues | Comments or suggestions to the community leaders | Making comments to people around you | Making comments through the media | Advice and suggestions to the government | Present a petition |
|---|---|---|---|---|---|---|
| Total | 33.4 | 43.5 | 60.8 | 13.2 | 12.5 | 15.1 |
| under age of 30 | 30.2 | 39.6 | 59.8 | 17.5 | 16 | 18.9 |
| 30–39 years old | 30.9 | 41.9 | 61.7 | 12.2 | 11.9 | 14.7 |
| 40–49 years old | 39.6 | 49.4 | 62.9 | 8.7 | 9 | 11.1 |
| over 50 years old | 41.8 | 51.3 | 58.7 | 9.5 | 8.1 | 11.1 |

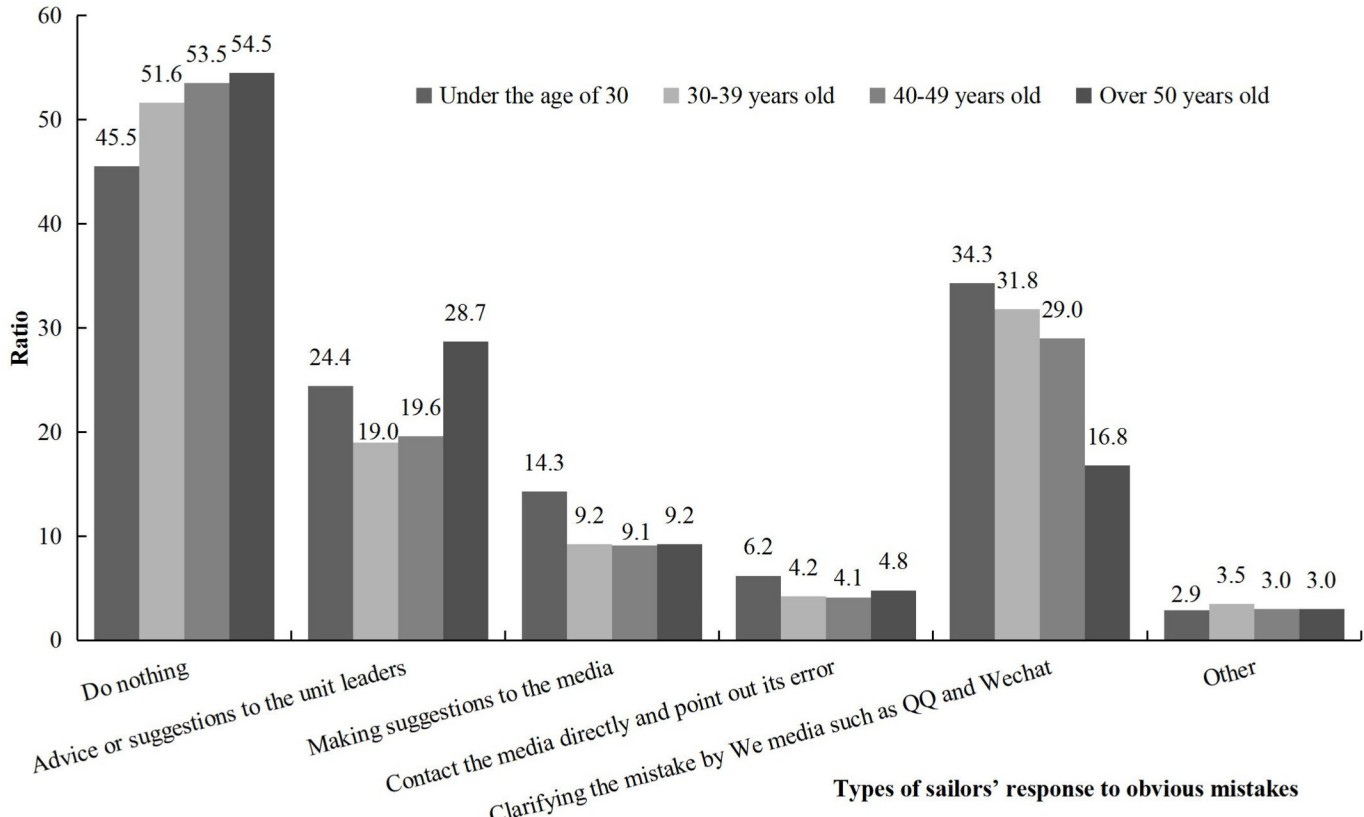

**Fig 4. Chinese merchant sailors' response to obvious mistakes in shipping- related information in the media by age group.**

information across age groups. Generally speaking, among the sailors who responded to this issue positively, most preferred an indirect approach (such as using social media). Comparatively, merchant sailors under 30 years old tended to report intention to actively clarify the mistake more often than their elders.

### 3.3 Evaluation and expectations of social participation

We also examined sailors' evaluation of current the social participation environment, including the channel smoothness of social participation, the influence of relevant social organizations, and their expectation of services provided by social organizations.

We found that only 19.1% of the sample thought that current channels for social participation were smooth (4.5% thought it was "very smooth", and 14.6% thought it was "smooth"); 53.6% of them thought it was not smooth (including 32.9% reporting "unsmooth" and 20.7% reporting "completely unsmooth"), with the remaining 27.3% expressing "Not sure". A $\chi^2$ test suggested significant differences among the responses. The majority of sailors thought the current social participation channels were not smooth, $\chi^2$ (2, $N = 6694$) = 16.78, $p < 0.01$.

The evaluation of the smoothness of social participation across age groups was similar (Fig 5). The proportion of sailors who replied "not sure" increased with sailors' age, with the highest proportion of sailors aged 30–39 indicating that current channels for social participation were not smooth (56.6%).

We investigated sailors' assessment of the influence of related social organizations, including the influence of the labor union related to Chinese seafarers and the NGO related to

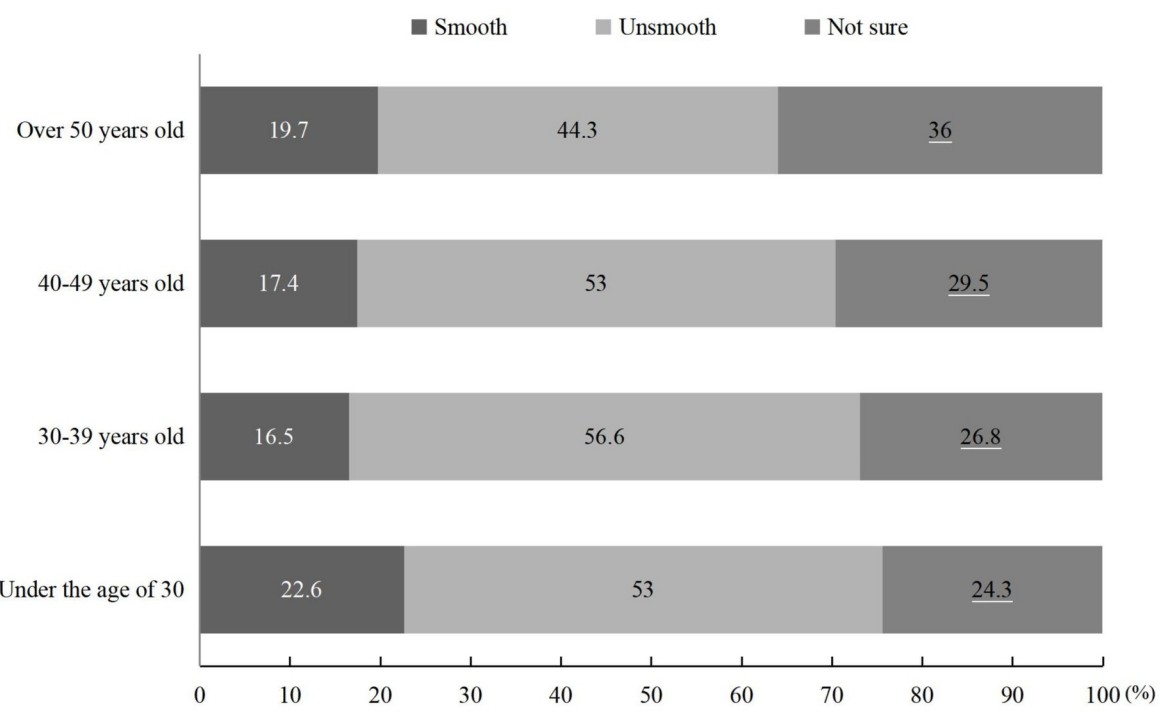

**Fig 5. Chinese merchant sailors' evaluation of channel fluency for social participation across age groups (%).**

navigation. The results showed that only 11.7% of the sample assessed that labor union related to Chinese seafarers had influence (including 2.6% sailors who considered it had "great influence" and 9.1% who considered that it had "influence"), while 51.3% thought that it had only a weak influence, and 26.4% reported that it had "no influence". Finally, 10.6% reported that they were "not sure". Similarly, only 13.5% of sailors thought that the NGO related to navigation had influence, while 50.2% of them believed that it had weak influence, and 23.6% thought that it had "no influence". Finally, 12.7% reported that they were "not sure".

Further we investigated sailor's understanding of the role of those two related social organizations. Our results showed that only 1.3% of them reported that they "Know very well" about the labor union related to Chinese seafarers, 15.6% of them reported they "know", while 83.1% reported they were "unknown". Similarly, only 1.2% of the sailors reported they "Know very well" about the NGO related to navigation, and 10.7% of them reported they "know", while 88.2% of them thought they were "unknown" about this organization.

We then examined sailors' assessment of labor union related to Chinese seafarers and the NGO related to navigation in relation to their degree of understanding of these bodies. As shown in Fig 6, different degrees of understanding of the labor union related to Chinese seafarers had different evaluations of its influence. Specifically, 43.5% of sailors who didn't know about labor union related to Chinese seafarers thought its influence was weak, and 37.5% of them reported that it was "influential", while among the sailors who knew the labor union related to Chinese seafarers well, only 33.3% of them thought its influence was weak, and 43.2% reported that they considered it to be "influential". A $\chi^2$ test suggested that merchant sailors who knew the NGO related to navigation better considered that its influence was weak, $\chi^2 (1, N = 6694) = 8.722, p < 0.05$.

Finally, we examined merchant sailors' expectations of the services provided by social organizations (Fig 7). They reported higher expectations of services in terms of protection of rights

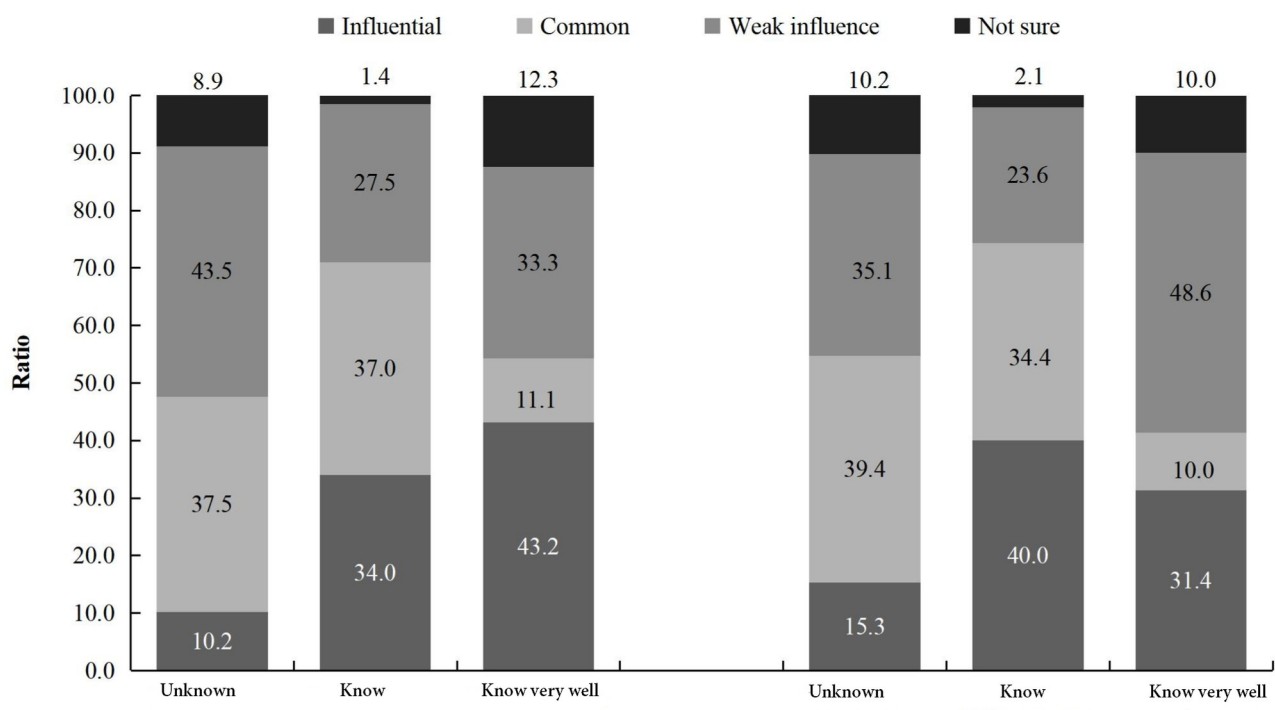

**Fig 6. Comments on the influence of the labor union related to Chinese seafarers and the NGO related to navigation by the sailors with different degrees of understanding of these two organizations (%).**

(49.9%), providing information and technology (49.8%), and providing employment opportunity (48.6%), but lower expectations were reported for services in funding research (10.7%)

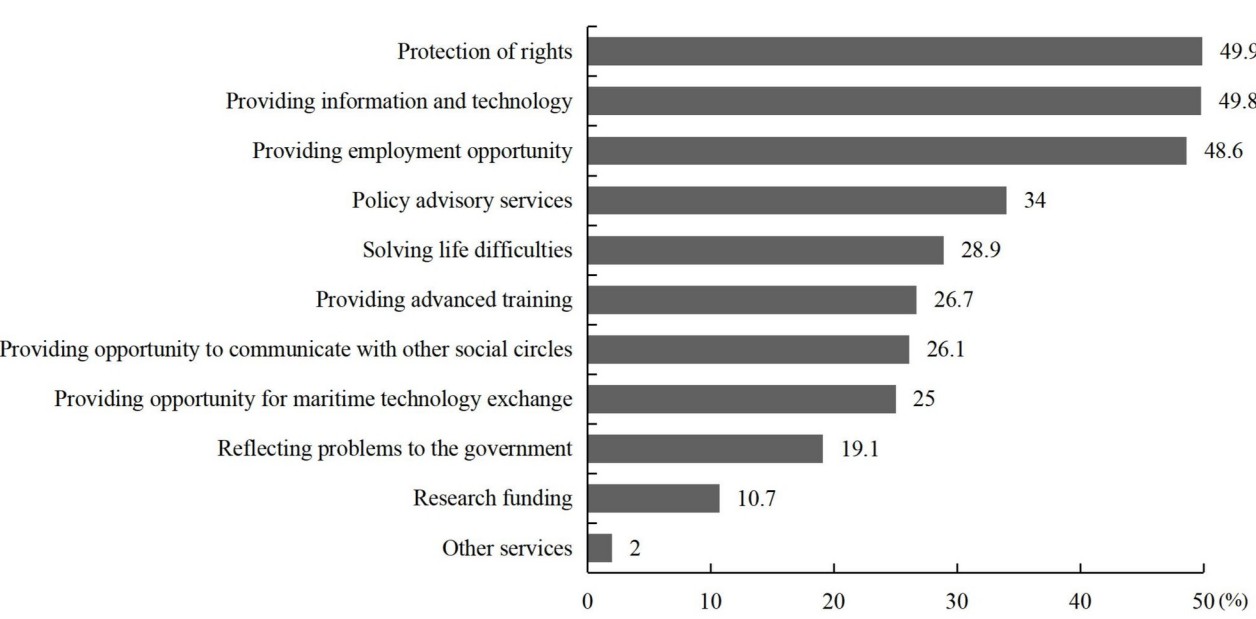

**Fig 7. Sailors' expectations for the provision of services from social organizations (%).**

## 4. Discussion

The results of a survey of 7,296 merchant sailors showed that most were willing to participate in social affairs, while they were not optimistic about the current state of social participation opportunities, but they had high expectations for the services provided by social organizations.

### 4.1 Willingness to participate in and reality of social participation

We found that most Chinese sailors reported being concerned about national policies and also being willing to participate in public affairs, indicating that they have a positive attitude of it. However, their actual social participation rate was less positive. Few reported having joined relevant social organizations: only 19.5% of the sample were members of the labor union related to Chinese seafarers and 12.6% were members of NGO related to navigation. This proportion was much lower than that found in a survey of 1000 international sailors carried out by the International Transport Workers' Federation (ITF) Seafarers' Trust [29], in which showed more than 50% of respondents reported being members of their national sailors' union. Over half of participants reported never having participated in public affairs in our sample, except for making comments to those around them, and when faced with an obvious mistake in shipping-related information in the media, half stated that they would choose to ignore it instead of taking a related action.

Several factors may explain the low social participation among Chinese merchant sailors. First, the reality of long-term offshore necessarily limits the degree to which sailors can participate in public affairs on shore, especially those who move cargo between nations would work longer on board. A previous study [29] of sailors reported that most of its respondents reported no access to the internet while at sea, and where there was access, it was expensive. Lack of information about shore affairs and lack of communication make it difficult for merchant sailors to participate in social affairs. Second, channels for social participation in China generally are not very smooth at the moment, which could reduce enthusiasm for social participation among all populations, including sailors. More than half of our respondents reported that current channels for social participation are not very smooth, which indicates that much remains to do to improve the smoothness of sailors' social participation channels. Finally, the lack of awareness of social participation among Chinese sailors may also be part of the reason. Unlike in Western culture [30], collectively oriented Chinese society allows Chinese people, including Chinese merchant sailors, to rely on the government to take care of social affairs instead of participating in public affairs themselves [31, 32], which indicates that much effort is needed to enhance awareness of social participation.

We found slight differences in social participation between age groups. Younger sailors, especially those who were under 30 years old, reported greater willingness to participate public affairs and indicated that they would respond more actively to incorrect shipping-related information, indicating a more positive attitude toward social participation than their elders. This reflects a positive trend that should continue into the future as the young sailors should carry their willingness and active participation in social affairs into the future and improve the social participation of sailors over time.

### 4.2 Evaluation and expectations of social organizations

Our study examined social organizations in relation to social participation as well, responding to previous research results that organizational support could improve the quality of sailors' life [33, 34]. However, we found that the labor union related to Chinese seafarers and NGO related to navigation, which both have a close relationship with Chinese sailors, did not seem to play the expected role among sailors, similar results in Guo's [35] study which focused on Chinese social organizations. Most of our sample reported not knowing the role of these

associations well. The respondents' evaluation of these organizations were mostly negative, but when analyzed by sailors level of understanding of the organizations, there were interesting differences between the labor union related to Chinese seafarers and NGO related to navigation. Specifically, sailors who knew the labor union related to Chinese seafarers well, tended to have a positive assessment, but the more the sailors knew about the NGO related to navigation, the less likely they were to believe it was influential. This finding offers valuable insight into the role social organizations play in sailors' social participation. First of all, social organizations need to improve their influence in sailors' groups, attract more sailors to join in and expand their popularity. The second and most important point, social organization need to provide effective services according to the needs of sailors. Regarding sailors' expectation for service, we also made a corresponding investigation. Our results showed that Chinese merchant sailors had a range of high expectations for service, including protection of rights, information and technology services, and employment services, which could provide direction for social organizations as they plan their initiatives.

## 4.3 Implications and limitations

Our results have implications for the improvement of social participation among Chinese merchant sailors. Firstly, active communication with the outside world is a basic element of social participation among sailors, whether at sea or on shore leave. Chinese merchant sailors should be encouraged to keep in touch with society at large, although this may prove difficult due to the separateness of their onboard and shorebound lives. However, the current development of satellite communications may help sailors easily obtain information on shore and communicate with outside world. Meanwhile, shipping companies can create conditions to increase the opportunities for sailors to participate in company affairs. By empowering sailors, they can obtain more resources and a stronger sense of control and professional belonging, which will effectively promote sailors' social participation.

Secondly, greater effort is needed to improve sailors' awareness of social participation. The government should improve the cognitive level of public participation, optimize social participation in the system environment, broaden the ways in which individuals can participate, promote better policies for social participation, and promote the civil rights for all citizens, including sailors. Meanwhile, the presence of a smooth channel for participation is key for sailors to take part. The development of networking technology had made it possible to participate in social affairs in novel ways [36]. In recent years, China's online means of political participation, government feedback hotlines, and other services have provided new ways for the masses to participate in public affairs [37], and these would naturally also benefit Chinese sailors.

Furthermore, social organizations must play a more prominent role in sailors' social participation. China ratified the ILO Maritime Labour Convention, 2006 (MLC, 2006) on November 12, 2015, which stipulated minimum requirements for employment, working conditions, the job environment, and the occupational health and safety protections of sailors. The MLC, 2006 has had a major impact on the working and living conditions of Chinese sailors, including their social participation. Meanwhile, national organizations, such as the labor union related to Chinese seafarers and NGO related to navigation, which have closest relationship to sailors' work, can work to attract sailors to join, by strengthening publicity, let more sailors know about these social organizations and the nature of their work. What's more, social organizations should provide practical support for sailors' specific needs, strive for more rights and interests for sailors, and solve practical obstacles for sailors [29].

Although this investigation was conducted with appropriate scientific rigor, it nevertheless had limitations. First, we relied on self-reported data, and our results may feature social

desirability bias. We assured participants that we would keep research findings confidential and anonymous, but they may have nevertheless exaggerated or altered their responses to comply with societal expectations [38]. Second, our sample was overbalanced to younger sailors and included few sailors 50 years old or older, which may affect its representativeness. Third, we did investigate the current state of social participation among sailors nationwide, but we could not determine the specific reasons for low social participation due to the shortcomings of the questionnaire method. Future study should adopt qualitative research methods such as interviews to supplement survey results.

## 5. Conclusion

Our study found that Chinese sailors are very willing to engage in social participation, although they did not exhibit high levels of social participation. We also found that most sailors' assessment of the current social participation environment was negative, and they expected certain actions from social organizations, including protection of their rights, the provision of information and technology, and providing employment opportunities, among other benefits.

Due to the lack of previous evidence, this study is the first exploratory investigation of the status quo and existing problems of Chinese sailors' social participation. Future research would integrate qualitative and quantitative research methods, further discuss in-depth internal and external factors that affect Chinese sailors' social participation, as well as optimizing strategies to improve the work and living conditions of sailors and enhance the sense of professional honor of Chinese sailors.

## Supporting information

**S1 File. Data underlying the findings.**
(DOCX)

**S2 File. Questionnaire in study.**
(DOCX)

## Acknowledgments

We thank the merchant sailors for their participation. We also thank 13 national maritime bureaus for their assistance with this study. Finally, we acknowledge Wang Qun, Zhu Yanhong, Yang Lianren and Meng Chunsheng for serving as research assistants for the study.

## Author Contributions

**Conceptualization:** Huarong Wang, Licheng Shi, Jiajun Dai.

**Data curation:** Huarong Wang, Licheng Shi, Jiali Wang, Lvqing Miao.

**Project administration:** Jiajun Dai.

**Supervision:** Jiajun Dai.

**Visualization:** Yuheng He.

**Writing – original draft:** Huarong Wang, Yuheng He.

**Writing – review & editing:** Huarong Wang.

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
