## [Decision Letter · Decision Letter 0]

5 Oct 2020

PONE-D-20-23544

Willingness to engage in and current status of social participation among Chinese merchant sailors

PLOS ONE

Dear Dr. Wang,

Thank you for submitting your manuscript to PLOS ONE. After careful consideration, we feel that it has merit but does not fully meet PLOS ONE’s publication criteria as it currently stands. Therefore, we invite you to submit a revised version of the manuscript that addresses the points raised during the review process.

Please read the comments of the reviewers carefully and try to follow them. Particularly the ones related to the PLOS Data policy.

We look forward to receiving your revised manuscript.

Kind regards,

Isabel Novo-Cortí

Academic Editor

PLOS ONE

Journal Requirements:

2. Please include additional information regarding the survey or questionnaire used in the study and ensure that you have provided sufficient details that others could replicate the analyses.

For instance, if you developed a questionnaire as part of this study and it is not under a copyright more restrictive than CC-BY, please include a copy, in both the original language and English, as Supporting Information.

3. Thank you for including your ethics statement:

'All participates provided informed consent to participate in the study. Approval for the research was obtained from the Nantong University Academic Ethics Committee prior to the study. Participants were guaranteed anonymity and that their answers would remain confidential. The study took about 10 to 15 minutes to complete for each participant.'

a. Please provide additional details regarding participant consent.

In the ethics statement in the Methods and online submission information, please ensure that you have specified what type you obtained (for instance, written or verbal, and if verbal, how it was documented and witnessed).

If your study included minors, state whether you obtained consent from parents or guardians.

If the need for consent was waived by the ethics committee, please include this information.

Additional Editor Comments:

Please read the comments of the reviewers carefully and try to follow them. Particularly the ones related to the PLOS Data policy indicated by the reviewer 1 (which requires authors to make all data underlying the findings described in their manuscript fully available without restriction, with rare exception (please refer to the Data Availability Statement in the manuscript PDF file). The data should be provided as part of the manuscript or its supporting information, or deposited to a public repository. For example, in addition to summary statistics, the data points behind means, medians and variance measures should be available. If there are restrictions on publicly sharing data—e.g. participant privacy or use of data from a third party—those must be specified).

Reviewers' comments:

Reviewer's Responses to Questions

**Comments to the Author**

1. Is the manuscript technically sound, and do the data support the conclusions?

Reviewer #1: Partly

Reviewer #2: Partly

2. Has the statistical analysis been performed appropriately and rigorously? 

Reviewer #1: Yes

Reviewer #2: N/A

3. Have the authors made all data underlying the findings in their manuscript fully available?

Reviewer #1: No

Reviewer #2: Yes

4. Is the manuscript presented in an intelligible fashion and written in standard English?

Reviewer #1: Yes

Reviewer #2: Yes

5. Review Comments to the Author

Reviewer #1: Having reviewed this research article, in my opinion, it satisfies the following criteria:

1. The research meets all applicable standards for the ethics of experimentation and the integrity of the research.

2. The article adheres to the appropriate reporting guidelines and standards for data availability.

3. The study presents the results of an original investigation.

4. The article is presented in an intelligible manner and is written in standard English.

5. The article is technically sound, supported by well-conducted statistical analysis.

6. The research methods described in section 2 of this article, using statistical analyzes, are carried out with a high technical level and are described in sufficient detail and rigorously.

7. The results described in section 3 of this article contain many data that provide much information, not published elsewhere.

Authors should be instructed to carry out a “Minor Review” of:

8. The conclusions of section 5 of this article, which should explain whether or not the hypotheses raised in this research are confirmed and propose future lines of research that expand and improve the data obtained in this article.

9. The authorizations granted, given that the data obtained in this investigation, are not completely available, applying some restrictions, which may be within the exceptions contemplated by the PLOS policy, given the peculiarities of China.

Reviewer #2: Dear Authors,

Your research is of interest and it could have some major implications.

That's why it's very important to emphasize the practical implications and to suggest some measures to achieve the proposed objectives:

- lines 341-343 - how do you suggest to encourage to keep in touch?

- lines 344-347 - by which means government could improve awareness?

- lines 359-362 - how LUCS and CIN could attract sailors to join?

Kind regards,

6. PLOS authors have the option to publish the peer review history of their article (what does this mean?). If published, this will include your full peer review and any attached files.

Reviewer #1: **Yes: **JOSÉ ÁNGEL FRAGUELA-FORMOSO

Reviewer #2: No

---

## [Author Response · Author response to Decision Letter 0]

1 Nov 2020

Dr. Isabel Novo-Cortí

em@editorialmanager.com

November 2, 2020

Dear Dr. Novo- Cortí,

Thanks to you and the reviewers for your consideration of our manuscript PONE-D-20-23544, titled “Willingness to engage in and current status of social participation among Chinese merchant sailors”. We were pleased to see the generally positive reviews and are happy to revise the paper in response to the comments. We believe the manuscript is improved as a result of the peer review.

Journal Requirements:

 Thank you, we’ve checked our manuscript fully and make sure it meets PLOS ONE’s style requirements.

2. Please include additional information regarding the survey or questionnaire used in the study and ensure that you have provided sufficient details that others could replicate the analyses.

For instance, if you developed a questionnaire as part of this study and it is not under a copyright more restrictive than CC-BY, please include a copy, in both the original language and English, as Supporting Information.

   Thank you. We’ve added the questionnaire (including both Chinese version and English version) in the additional information.

3. Thank you for including your ethics statement:

'All participates provided informed consent to participate in the study. Approval for the research was obtained from the Nantong University Academic Ethics Committee prior to the study. Participants were guaranteed anonymity and that their answers would remain confidential. The study took about 10 to 15 minutes to complete for each participant.'

a. Please provide additional details regarding participant consent.

In the ethics statement in the Methods and online submission information, please ensure that you have specified what type you obtained (for instance, written or verbal, and if verbal, how it was documented and witnessed).

If your study included minors, state whether you obtained consent from parents or guardians.

If the need for consent was waived by the ethics committee, please include this information.

 Thank you for your suggestion. 

 Regarding the informed consent of the participants, considering the large number of sailors that need to be investigated in our research, we requested the help of the National Maritime Safety Administration, and with its support, arranged its 13 local maritime safety bureaus to help us to organize local shipping companies to participate survey. The shipping companies informed their sailors of this information about participating in the investigation. Our research assistants introduced to the sailors the purpose and specific requirements of the investigation, and clearly informed the principle of voluntary participation in the research.Only the sailors who agreed to participate in the survey stayed, and the research assistants would guide him/them to fill out the questionnaire, while those who didn’t agree to participate in the study can leave without filling in the questionnaire. Considering the large number of people participating in the survey, this study adopted the method of verbal informed consent.

  Thank you for your suggestion.We’ve added it.

Additional Editor Comments:

Please read the comments of the reviewers carefully and try to follow them. Particularly the ones related to the PLOS Data policy indicated by the reviewer 1 (which requires authors to make all data underlying the findings described in their manuscript fully available without restriction, with rare exception (please refer to the Data Availability Statement in the manuscript PDF file). The data should be provided as part of the manuscript or its supporting information, or deposited to a public repository. For example, in addition to summary statistics, the data points behind means, medians and variance measures should be available. If there are restrictions on publicly sharing data—e.g. participant privacy or use of data from a third party—those must be specified).

Thank you for your valuable suggestion, we’ve added all data underlying the findings described in the manuscript as supporting information.

Reviewers' comments:

Reviewer's Responses to Questions

Comments to the Author

1. Is the manuscript technically sound, and do the data support the conclusions?

Reviewer #1: Partly

Reviewer #2: Partly

2. Has the statistical analysis been performed appropriately and rigorously?

Reviewer #1: Yes

Reviewer #2: N/A

3. Have the authors made all data underlying the findings in their manuscript fully available?

Reviewer #1: No

Reviewer #2: Yes

4. Is the manuscript presented in an intelligible fashion and written in standard English?

Reviewer #1: Yes

Reviewer #2: Yes

5. Review Comments to the Author

Reviewer #1: Having reviewed this research article, in my opinion, it satisfies the following criteria: 

1. The research meets all applicable standards for the ethics of experimentation and the integrity of the research.

2. The article adheres to the appropriate reporting guidelines and standards for data availability.

3. The study presents the results of an original investigation.

4. The article is presented in an intelligible manner and is written in standard English.

5. The article is technically sound, supported by well-conducted statistical analysis.

6. The research methods described in section 2 of this article, using statistical analyzes, are carried out with a high technical level and are described in sufficient detail and rigorously.

7. The results described in section 3 of this article contain many data that provide much information, not published elsewhere.

 Thanks for your positive feedback– no response required.

Authors should be instructed to carry out a “Minor Review” of:

8. The conclusions of section 5 of this article, which should explain whether or not the hypotheses raised in this research are confirmed and propose future lines of research that expand and improve the data obtained in this article.

 Thank you for your valuable suggestions. Due to the lack of previous evidence, we designed an exploratory study and didn’t propose any hypotheses(we mentioned in section 1: introduction). However, our research provides a meaningful empirical basis for future related research, and we added this part of information in the conclusion section.

9. The authorizations granted, given that the data obtained in this investigation, are not completely available, applying some restrictions, which may be within the exceptions contemplated by the PLOS policy, given the peculiarities of China.

 Thank you for your suggestion, we’ve provided all the data as supporting information.

Reviewer #2: Dear Authors,

Your research is of interest and it could have some major implications.

That's why it's very important to emphasize the practical implications and to suggest some measures to achieve the proposed objectives:

- lines 341-343 - how do you suggest to encourage to keep in touch?

 Good question. We’ve added more information in the manuscript about the suggestion to encourage sailors to keep in touch with society. 

- lines 344-347 - by which means government could improve awareness?

 Thank you for your valuable question. we mentioned several ways that the government can take to improve people’s awareness of social participation in our manuscript, including improving the cognitive level of public participation, optimizing social participation in the system environment, broadening the ways in which individuals can participate, promoting better policies for social participation, and promoting the civil rights for all citizens(In line 356-365).

- lines 359-362 - how LUCS and CIN could attract sailors to join?

 Good question. We’ve added more detailed information to the manuscript.

In summary, we have made substantial revisions in response to all reviews. We believe the manuscript is improved as a result and thank the reviewers and editor for their suggestions. We look forward to hearing your editorial decision in due course.

 Sincerely,

 Huarong Wang, PhD

---

## [Editor Report · Decision Letter 1]

11 Nov 2020

Willingness to engage in and current status of social participation among Chinese merchant sailors

PONE-D-20-23544R1

Dear Dr. Wang,

We’re pleased to inform you that your manuscript has been judged scientifically suitable for publication and will be formally accepted for publication once it meets all outstanding technical requirements.

Kind regards,

Isabel Novo-Cortí

Academic Editor

PLOS ONE

---

## [Editor Report · Acceptance letter]

13 Nov 2020

PONE-D-20-23544R1 

Willingness to engage in and current status of social participation among Chinese merchant sailors 

Dear Dr. Wang:

I'm pleased to inform you that your manuscript has been deemed suitable for publication in PLOS ONE. Congratulations! Your manuscript is now with our production department. 

Kind regards, 

on behalf of

Dr. Isabel Novo-Cortí 

Academic Editor

PLOS ONE